# Perception of Health, Mistrust, Anxiety, and Indecision in a Group of Italians Vaccinated against COVID-19

**DOI:** 10.3390/vaccines9060612

**Published:** 2021-06-07

**Authors:** Giuseppina Moccia, Luna Carpinelli, Giulia Savarese, Anna Borrelli, Giovanni Boccia, Oriana Motta, Mario Capunzo, Francesco De Caro

**Affiliations:** 1Department of Medicine, Surgery and Dentistry “Scuola Medica Salernitana”, University of Salerno, 84081 Baronissi, Italy; gmoccia@unisa.it (G.M.); lcarpinelli@unisa.it (L.C.); gboccia@unisa.it (G.B.); omotta@unisa.it (O.M.); mcapunzo@unisa.it (M.C.); fdecaro@unisa.it (F.D.C.); 2A.U.O. “San Giovanni di Dio e Ruggi d’Aragona”, 84121 Salerno, Italy; direzione.sanitaria@sangiovannieruggi.it

**Keywords:** COVID-19, vaccination, psychological perception

## Abstract

The objectives of this study were to evaluate the psychological factors of health perception, mistrust, anxiety, fear, and indecision of Italians vaccinated against COVID-19, and conduct an analysis of the relationships between these factors and other variables: sex, vaccine priority ministerial categories, and the type and dose of vaccine. The participants included 1564 subjects who joined the vaccination campaign at the COVID-19 Vaccination Center in Salerno, Italy. A survey was conducted in the reference period March–April 2021 using a brief anamnestic questionnaire. In addition, the following standardized scales were used: the State–Trait Anxiety Inventory (STAI-Y) and the Short Form Health Survey (SF-12). The results showed that, in terms of the type of vaccine received, the interviewees felt more confident in having received the Comirnaty (Pfizer-BioNTech, 23.5%) and Vaxzevria (AstraZeneca, 18.6%) vaccines—feeling less tense (2.1%; Vaxzevria (AstraZeneca) = 3.2%), frightened (1%; Vaxzevria (AstraZeneca) = 1.4%), not at all nervous (61.1%; Vaxzevria (AstraZeneca), 43.6%), and not at all/undecided (67.9%; Vaxzevria (AstraZeneca), 58.6%). Regarding the mood and psychological states considered at the different vaccine administration times, other important differences emerged as the interviewees reported higher levels of tension, nervousness, and fear during the first phase of vaccine administration. Specifically, 40.7% (second dose, 32.7%) felt somewhat tense at the first dose, 26.4% felt frightened (second dose, 21.8%), and 33.8% felt nervous (second dose, 26.8%). The perceived state of health also increased at the end of the vaccination cycle, as, at the second dose, 15.4% of the sample reported an evaluation of “excellent” (first dose, 12.4%).

## 1. Introduction

The COVID-19 vaccines currently represent an effective weapon with which to face and defeat the ongoing pandemic. The fears causing increased vaccine resistance on the part of patients are based on both mild and severe reactions following vaccination and the time difference between doses [1]. It is evident that this resistance or hesitancy can undermine the success of the current vaccination campaign, even if it only occurs in a small group of the population. These fears are also present in healthcare personnel, despite the presence of corresponding psychological antecedents such as trust, complacency, and sense of responsibility, and the mediating effects of work stress. Karafillakis et al. [2] found that a number of healthcare professionals are hesitant to receive a vaccination, although they are often cited as the most reliable source of vaccine information. Kwok et al. [2] interviewed a group of professional nurses and identified higher willingness to vaccinate in specific categories: the youngest, those most confident in institutions, and those with increased collective responsibility and higher work stress regarding the management of patients with COVID-19. The most important concern is fear of side effects from the vaccine, as well as a strong distrust in pharmaceutical companies due to alleged perceived financial interest and a lack of clear communication about the side effects. Healthcare professionals have the potential to influence patient vaccination adoption and are critical for improving vaccination confidence [2].

Reiter et al. [3] surveyed U.S. adults, finding that 69% were willing to receive a COVID-19 vaccine, and that this percentage was higher in those who reported higher levels of perception of the possibility of contracting COVID-19 or awareness of the perceived severity of the infection and the effectiveness of the vaccines. For Manning et al. [4], the main reasons for vaccine refusal are related to information regarding the safety of the vaccine and the side effects. In a study by Detoc et al. [5], between 75% and 48% of respondents declared a propensity to accept vaccination or participate in a clinical trial against COVID-19. Borriello et al. [1] inferred that people would pay to receive a vaccine, but only under certain conditions: immediacy, effectiveness, and mild side effects.

It is, therefore, clear that identifying the psychological roots of vaccination hesitancy is essential for achieving high vaccination rates, as well as guiding the creation of educational campaigns to increase compliance [6].

In Israel, the country with the highest vaccination rate, Palgi et al. [7] examined psychiatric comorbidities and attitudes toward the vaccine among individuals who had already been vaccinated, finding higher levels of vaccination hesitation in patients with depression and pre-traumatic stress, and finding that failure to receive the vaccine can triple the risk of anxiety.

Byrne et al. [8] found that attitudes, beliefs, and emotions related to the COVID-19 disease and vaccine influence the intention to receive vaccination. Murphy et al. [9] concluded that having confidence in the safety of vaccines is associated with a significantly higher intention of accepting the vaccine. This is in line with other studies reporting that trust is a determinant of vaccine uptake [9], indicating the need for educational and communication strategies aiming to increase trust among people with higher levels of vaccine skepticism [10]. Furthermore, in Italy, Graffigna et al. [11] verified that the perceived severity of contracting COVID-19 and general vaccine attitudes affect confidence in vaccination.

Ward et al. [12] found that attitudes toward the vaccine are significantly correlated with orientation and engagement within the political system. In this regard, Salali et al. [13] demonstrated that individuals’ lack of confidence in the efficacy and safety of vaccines is, in part, influenced by circulating political conspiracy theories about COVID-19 vaccination conveyed by the media. Trujillo et al. [10] also inferred that a lower sense of collective responsibility is associated with a lower intention to vaccinate against COVID-19. This result is in line with other studies confirming the role of this psychological factor [14,15].

In the months since the vaccination campaign began, a series of studies have also documented the impact of the psychological factors of vaccine efficacy on the immune system. For example, Madison et al. [16] showed that stress, depression, loneliness, and poor health behaviors can impair the immune system’s response to vaccines, and that this effect may be greater in vulnerable groups, such as the elderly.

To define elements of the territorial context, we report some data. The website of the Italian government [17], with reference to the month of May 2021, shows that 4,217,821 people have been infected since the beginning of the pandemic, of which 419,000 were from the region from which we collected the data, Campania [18,19]. The total number of vaccines administered in Italy is currently 35,578,293, of which 66.3% are of the Comirnaty (Pfizer/BioNTech) type, 20.2% Vaxzevria (AstraZeneca) type, 10.1% Moderna type, and 4.4% Jansenn type. In Campania, 3,349,658 vaccines were administered, 96.2% of the doses were delivered. In comparison, the Province of Salerno, home to the hospital for the patients interviewed, has a population of 1,081,000 inhabitants; 490,000 participated in the vaccination campaign (registration on the regional booking platform “Sinfonia”) and, of these, 457,000 have already been vaccinated, or 42.2% of the total population. Here, only Comirnaty (Pfizer/BioNTech) and Vaxzevria (AstraZeneca) vaccines were administered. The other vaccines are administered by the ASLs and not by hospitals.

The third pharmacovigilance report on COVID-19 vaccines by Aifa [20], the Italian national pharmacovigilance agency, highlights a reported rate of 510 adverse reactions per 100,000 doses. A total of 92.7% refers to non-serious events, which are completely resolved; serious reports correspond to 7.1% of the total, with a rate of 36 serious events per 100 thousand doses administered, regardless of the type of vaccine, the dose (first or second) and the possible causal role of the vaccination. Most of the reports are related to the Comirnaty (Pfizer-BioNTech) vaccine (81%), which are, to date, the most used in the vaccination campaign (68% of the doses administered), with an increase in reports for the Vaxzevria (Astrazeneca) vaccine (17%), which underlines the Aifa report, following the increase in the use of this vaccine (27% of the doses administered). The reports relating to the Moderna vaccine represent 2% of the total and are proportional to the more limited number of doses administered (5%). There are no data showing differences in terms of age or ministerial categories.

Given these theoretical-descriptive premises, the objectives of this study were: (1) to evaluate the psychological factors of health perception, mistrust, anxiety, fear, and indecision of Italians vaccinated against COVID-19; (2) to conduct an analysis of the relationships between these factors and the variables, and to investigate and understand whether gender, ministerial priority categories for vaccines, type and dose of vaccine have given different results. This second objective interests us due to the absence of Italian studies on psychosocial variables that can influence the acceptance of vaccination.

## 2. Methods

### 2.1. Participants

A total of 1564 subjects (54.2% women; Mean age = 56.29; SD = 55.369) who joined the vaccination campaign at the COVID Vaccination Center of the Azienda Ospedaliera Universitaria (A.O.U.) “San Giovanni di Dio and Ruggi d’Aragona”, in Salerno, Italy, participated in the administration of the questionnaire. As Table 1 shows, 65.8% of the participants had received the first vaccine dose, while 34.2% had received both doses (first and second for the vaccine administration process); 75.7% received the Comirnaty vaccine (Pfizer-BioNTech) and 24.3% received the Vaxzevria (AstraZeneca) vaccine. At the end of the data collection period (April 2021), only the subjects who received the Comirnaty vaccine (Pfizer-BioNTech) had completed the vaccination course (1st and 2nd dose), whereas the second doses of Vaxzevria (AstraZeneca) were planned to be administered to subjects starting in June 2021.

The group of participants was also divided according to the priority ministerial categories in the initial phase of the vaccination campaign: 28.1% medical and health personnel; 10.6% administrative staff; 15.6% school staff; 3.1% university staff; 10.3% people with frailty; 0.4% people with disabilities; 1.1% law enforcement agencies; 28.8% people with advanced age (>80 years of age); 2% subjects aged between 70 and 79 years.

Of the total sample, 22.8% had chronic diseases, 8.8% had gait disturbances, 24.2% complained of joint pain, 7.3% suffered from diabetes, 26.4% suffered from hypertension, and 12.1% had heart disease.

Regarding the socio-economic status, in Italy the health system is free, and as such, this variable has not been investigated.

### 2.2. Instruments

A survey was administered in the reference period March–April 2021 using an ad hoc questionnaire, built based on the variables covered by the study. Specifically, brief questions were used, both anamnestic and related to the possible presence of syndromes and/or pathologies. In addition, the following standardized scales were used.

The State–Trait Anxiety Inventory (STAI-Y) [21] is a useful tool for detecting and measuring anxiety, consisting of 40 items, to which the subject must respond on a Likert scale in terms of intensity (from “not at all” to “very much so”). The items are grouped into two scales, focusing on how the subjects generally feel or what they feel in particular moments, to investigate (a) state anxiety, where anxiety is conceived as a particular experience, a feeling of insecurity or of helplessness in the face of perceived harm that can lead to either worry or flight and avoidance, and (b) trait anxiety, which is the tendency to perceive stressful situations as dangerous and threatening and to respond to various situations with different intensities. In the present study, Scale 1, which is specific to state anxiety, was examined.

The Short Form Health Survey (SF-12) [22] is a questionnaire that aims to investigate the perception of individuals’ psychophysical conditions. It is taken from a more extensive version, the SF-36, a multidimensional questionnaire that is divided into 36 items from eight dimensions (physical activity, role limitations due to physical health, emotional state, physical pain, perception of general health, vitality, social activities, and mental health). In the SF-12, the summary of the scores allows for the construction of two indices of the state of health: one concerning the physical state (Physical Component Summary (PCS) and the other concerning psychological state (Mental Component Summary (MCS)). The values of the synthetic indices vary from 10.5 to 69.7 for the PCS and from 7.4 to 72.1 for the MCS index, with higher values indicating better psychophysical health. Very low (roughly below 20 points) PCS scores correspond to a condition of “substantial limitations in self-care and physical, social and personal activity; severe physical pain; frequent fatigue; health is judged to be poor”. A low psychological health status (MCS) value highlights “frequent psychological distress; significant social and personal disability due to emotional problems; health is judged to be poor”.

### 2.3. Procedure

At the “San Giovanni di Dio e Ruggi d’Aragona” University Hospital, under resolution 493 (“COVID vaccination business plan—Phase 1”), the COVID Vaccination Center was created based on specific principles and organizational models for the innovative control and management of the epidemiological context of SARS-Cov-2. The Vaccinal Center of the Salerno Hospital was created at the University of Salerno’s Pole for Health Professions, a strategic point that was able to guarantee the total separation of people who needed to be vaccinated from normal hospital users. The organization of the spaces was designed to ensure maximum efficiency, from the moment of acceptance until leaving the post-vaccination observation room, in order to harmonize the vaccination process of each patient. The vaccination process begins for each patient with acceptance at the Single Booking Center, where the vaccinated, in compliance with the rules of social distancing, initiate the recognition and registration procedures on the regional platform. Subsequently, the vaccinator is directed to the entrance to the Center, where the health and social workers measure body temperature and hands are disinfected, and then the subjects are sorted between the various clinics, where the vaccine is administered.

The current organizational structure allows for the presence of a doctor and a nurse in each clinic: the doctor is responsible for supervising the vaccination session, granting eligibility for administration, and reporting and filling in reports of adverse reactions and administration of the vaccine; the nurse participates in the preparation of the patient, administers the vaccine as an alternative to the doctor, manages the computerized procedures for the registration of vaccinations on the regional platform, and prints the vaccination certificate.

After vaccine administration, the observation phase is scheduled in the post-vaccination room, which, according to protocol, is the largest room in the entire Center, guaranteeing interpersonal distancing and contingencies.

The questionnaire was administered in the post-vaccination room in which the subjects waited from 15 to 30 min (depending on the risk of adverse reactions due to pathologies and/or allergies). The subjects were provided an informed consent form to sign, where the purpose of the investigation was indicated. The time taken to complete the questionnaire was about 5 min by scanning the QRCode that referred to the form on Google Forms or, alternatively, in paper form. For the categories of people with frailty, disability, and/or advanced age, the method of administration with an interviewer was chosen to better facilitate the completion by the subject.

## 3. Statistical Analysis

Statistical analysis was carried out using IBM software SPSS vs. 23.0.

Both qualitative and descriptive analyses of the items based on the frequency of the subjects’ responses and correlation analyzes were conducted to verify the actual presence of significance among the variables. Through the descriptive analysis with the use of cross-tables and ANOVA, it was possible to verify the differences between the groups.

## 4. Results

In relation to the first objective, that is, to evaluate the psychological factors of the perception of health, mistrust, anxiety, fear, and indecision of Italians vaccinated against COVID-19, the sample of vaccinated subjects reported state-of-health perceptions within the norm. By analyzing the Physical Component Summary (PCS) and the Mental Component Summary (MCS) obtained from the SF-12, it was observed that the total sample reported scores that fell within the normative range |0–39|, which indicates a functional level of perception of one’s health, both physically and mentally (M = 1.00; SD = 1.000), and showed no differences between the other variables investigated, except in percentage terms between the phases of vaccine administration. The data showed an increase in the level of health perceived at the second vaccine dose, since 15.4% of the sample perceived their health as “excellent” compared with 12.4% at the first dose.

Considering the psychological constructs under investigation, we report a qualitative analysis of the related items on the SF-12 test (see Figure 1 and Figure 2). The percentages obtained for item #9, “calm and peaceful”, indicated a substantial difference between the sexes, as women reported lower values in the answers “all of the time” (10.9%) and “most of the time” (31.6%) compared with men (Table 2). According to the type of vaccine, among the subjects who received Comirnaty (Pfizer-BioNTech), there was a higher percentage of the answers “all of the time” (15.9%) and “most of the time” (36.3%) compared with those who received Vaxzevria (AstraZeneca) (Table 2). Different percentages were found between the first and second vaccine administration; 36% reported feelings of being calm and peaceful “most of the time” and 26.9% for “some of the time” at the first dose. At the end of the vaccination cycle, 18.3% of the subjects responded that they felt calm and peaceful all of the time (Table 2).

Comparing the percentage analysis between the categories adhering to the vaccination campaign, as shown in Table 3, we found that the law enforcement staff predominantly responded to the question regarding feeling calm and peaceful with “all of the time” (29.4%) and “most of the time” (52.9%) compared with the others, who fluctuated more in their answers, from “a good bit of the time” to “none of the time”. Specifically, 28.6% of people with frailty reported that they felt calm and peaceful “some of the time” and 10.9%, “a little of the time”; 36.7% of university staff and 26.8% of administrative staff reported “Some of the time”; 9.4% of those over eighty replied “a little of the time”.

Table 4 shows the responses to item #11 of the SF-12, which identifies feelings of discouragement and sadness. More specifically, women (36.9%) felt sad “some of the time” compared with 25.1% of men, of whom 41.9% reported “a little of the time” and 24.4% reported “none of the time”. In relation to the type of vaccine, 34.6% of the subjects who received Comirnaty (Pfizer-BioNTech) answered “a little of the time” against the 41.2% of those who received Vaxzevria (AstraZeneca). Regarding the administered dose, we observed a percentage increase in reference to the first dose, where 32.8% of subjects reported such moods “some of the time”. Considering the differences in the type of vaccine, it was interesting to assess whether there was mistrust towards Vaxzevria (AstraZeneca) after the alarmism in Italy and the rest of Europe, especially in the media, concerning the cases of thrombophilia resulting from the inoculation of the Vaxzevria (AstraZeneca) vaccine.

By concentrating on the qualitative analysis of the reference categories, Table 5 shows some important differences between them. Specifically, we note that the feelings of discouragement and sadness were more frequent in people with disabilities (50%), who reported experiencing them “a good bit of the time”; 31.2% of those over 80 and 44.4% of university staff responded with “some of the time”. The medical and health personnel (41.5%), administrative personnel (44%), school personnel (39%), and people with frailty (38.8%) responded with “a little of the time”. Only the police, on the whole, were more likely to respond with “none of the time” (66.7%).

We performed a qualitative analysis of the items of scale 1 of the STAI-Y that were found to be the most significant (*p* = 0.01) in reference to the stressful event “vaccination”. In this specific case, as shown in Table 6, we evaluated items related to the feeling of security and states of anxiety, tension, fear, nervousness, and indecision, looking for any differences between the sexes, type and dose of vaccine received, as well as any percentage differences in the answers obtained between the different categories under examination (see Figure 3). The results showed that, when investigating whether they feel secure with respect to vaccination, 78% of women responded with “moderately so”, compared with only 66.9% of men, who had a higher percentage (29.7%) in the answer “very much so” compared to women (16.2%). Considering the type of vaccine, 23.5% of those who received Comirnaty (Pfizer-BioNTech) responded with “very much so” (Vaxzevria (AstraZeneca) = 18.6%), and there were no obvious differences between the first and second received doses. States of tension, nervousness, fear, and indecision were more frequent in women, who reported higher percentages than men in the answers “somewhat” and “not at all”. Obvious differences were also observed for the type of vaccine inoculated, as those who vaccinated with Vaxzevria (AstraZeneca) reported higher states of tension (“very much so” = 3.2%; Comirnaty (Pfizer-BioNTech) = 2.1%), being frightened (“somewhat” = 30.4%; Comirnaty (Pfizer-BioNTech) = 23%), nervousness (“somewhat” = 43.9%; Comirnaty (Pfizer-BioNTech) = 27.5%), and indecision (“moderately so” = 9.5%; Comirnaty (Pfizer-BioNTech) = 6.1%). The state of tension that emerged to varying extent between the first and second dose administrations is also relevant: subjects reported a higher level of tension (“somewhat” = 40.7%; second dose = 32.7%) and nervousness (“somewhat” = 33.8%; second dose = 26.8%) in the first administration.

From Table 7, it is possible to verify the differences between the categories that participated in the vaccination campaign (see Figure 4). A total of 41.2% of the police force replied with “very much so” to the item “I feel secure”, followed by subjects aged between 70 and 79 years (31%), and elderly people (over 80 = 30.9%). Greater tension was present in medical and health personnel, of whom 13.3% responded to the item with “moderately so”, followed by the categories of people with frailty (12.5%), and school personnel (12.4%). The feeling of fear was more evident in the law enforcement vaccine priority ministerial categories, of whom 11.8% responded with “moderately so”, as did people with frailty (10.5%), and university staff (10.4%). Nervousness was higher in people with disabilities, 80% of whom responded with “somewhat”, as did 45.8% of university staff and 45% of school staff, people with frailty (39.3%), administrative staff (38.9%), and individuals aged between 70 and 79 years (38.9%). Regarding adherence to the vaccination campaign, the categories that reported the least indecision were those who opted for “not at all” as their answer to the item “I feel indecisive”, namely, people with disabilities (83.3%), people aged between 70 and 79 years (79.3%) and over 80 years (76.6%), and the police (76.5%). Categories such as university staff (38.8%), people with frailty (32.2%), and medical and health personnel (29.1%) answered “somewhat”.

Finally, with respect to the second objective, that is, to conduct a correlational analysis between the psychological factors and the variables investigated, an ANOVA was performed between the PCS and MCS components of the SF-12, using the cumulative average of the STAI-Y scale 1 scores and the variables considered. Vaccine type significantly correlated with sex (*p* = 0.079 **), PCS (*p* = 0.175 **), MCS (*p* = 0.151 **), and STAI-Y (*p* = 0.115 **). PCS correlated significantly with MCS (*p* = 0.504 **) and STAI-Y (*p* = 0.182 **). The same effect also emerged in the correlation between STAI-Y and age (*p* = −0.102 **), in which the anxiety index decreased as the reference age range decreased. For the categories, we found numerous significant correlations with both sex (*p* = 0.100 **), the type of vaccine (*p* = 0.352 **), and STAI-Y (*p* = 0.134 *), whereas the results showed negative correlations with other variables, such as age (*p* = −0.199 **), dose received (*p* = −0.293 **), PCS (*p* = −0.075 **), and MCS (*p* = −0.180 **). As shown in Table 8, very low (roughly below 20 points) PCS scores correspond to a condition of “substantial limitations in self-care and physical, social and personal activity; severe physical pain; frequent fatigue; health is judged to be poor”. A low psychological health status (MCS) value highlights “frequent psychological distress; significant social and personal disability due to emotional problems; health is judged to be poor”.

## 5. Discussion

Safety concerns, side effects, and rapid vaccine development have been cited as barriers to vaccination [23,24], and we think that they have influenced the manifestations of anxiety, fear, and insecurity in our interviewees, especially in relation to the type and dose of vaccine received. As demonstrated by the analysis of the results, a difference emerged for the type of vaccine received, as the interviewees felt more confident in receiving the Comirnaty (23.5%; Pfizer-BioNTech) vaccine, and (Vaxzevria (AstraZeneca) = 18.6%) vaccine, less tense (2.1%; Vaxzevria (AstraZeneca) = 3.2%), less frightened (1%; Vaxzevria (AstraZeneca) = 1.4%), “not at all” nervous (61.1%; Vaxzevria (AstraZeneca) = 43.6%), and “not at all” undecided (67.9%; Vaxzevria (AstraZeneca) = 58.6%). Regarding the mood and psychological states considered at the different vaccine administration times, other important differences emerged. The interviewees reported higher levels of tension, nervousness, and fear during the first phase of vaccine administration. Specifically, 40.7% (second dose = 32.7%) felt “somewhat” tense at the first dose, 26.4% frightened (second dose = 21.8%), and nervous (33.8%; second dose = 26.8%). Furthermore, the perceived state of health also increased at the end of the vaccination cycle, as 15.4% of the sample evaluated it as “excellent” at the second dose (first dose = 12.4%).

We observed that, with regard to vaccine confidence, our results are similar to those of the Coconel longitudinal study conducted by the Observatoire Régional de Santé Provence Alpes Côte d’Azur [25] and the results of surveys conducted in the United States [26]. Regarding the Coconel, we observed that men felt safer than women, as 16.2% of women responded that they “very much so” felt secure, compared with 29.7% of the men interviewed.

Older individuals were less hesitant and reported higher levels of confidence (30.9%) than other categories. This is probably due to the higher perceived risk of infection and development of serious disease in people with advanced age.

Similarly, medical and health personnel (19%) and law enforcement agencies (41.9%) reported feeling more secure with vaccination than the other categories considered, probably because these categories have a greater perceived risk of being infected, especially during the first wave, in which they were at the forefront, in addition to the sense of professional responsibility felt toward the citizens to whom they provide both health and safety services. This is in line with the related literature, as healthcare professionals were found to have the potential to influence patient vaccination adoption and are critical for improving vaccination confidence [27]. Widespread anxiety and psychological aspects caused by the pandemic have impacted health perception and factors related to vaccination [5,28,29,30]. It is possible that this is related to the myriad of information received, which serves as a double-edged sword: for some, the information helped ease anxiety and the stigma surrounding vaccination, and motivated them to accept the vaccine; conversely, this information also created many negative perceptions in the community about vaccination [31]. The literature [32] also suggests that the lack of confidence in the efficacy and safety of vaccines is partly influenced by circulating conspiracy theories about COVID-19 vaccination, and that a lower sense of collective responsibility is associated with a lower intention to get vaccinated. A cross-sectional survey with 1912 Chinese university students identified facilitating variables for the acceptance of vaccines: the lower socio-economic status, the female gender, greater perception of the risk of getting sick, and greater prosociality characteristics [33]. In Italy, a research group interviewed 735 university students, and it emerged that 86% said they wanted to be vaccinated. However, the researchers reflected on the fact that more than one in 10 students showed vaccination hesitation [34].

## 6. Conclusions

The results that emerged in the present study, combined with the research present in the literature, suggest that it is of fundamental importance to consider specific psychological determinants of vaccine acceptance [31]. In our work, we investigated various determinants, including health perception, mistrust, anxiety, fears, and indecision, highlighting how they were influenced by some variables, such as the type of vaccine administered, age, and whether respondents fell into a professional role involving the care and safety of other people.

## Figures and Tables

**Figure 1 vaccines-09-00612-f001:**
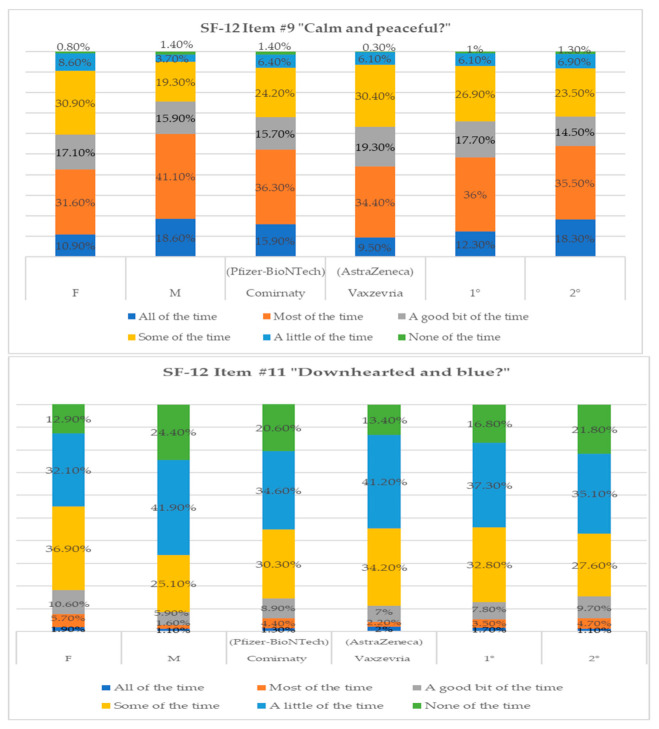
Percentage of response frequencies relative to items #9 and #11 of SF-12 divided by variable.

**Figure 2 vaccines-09-00612-f002:**
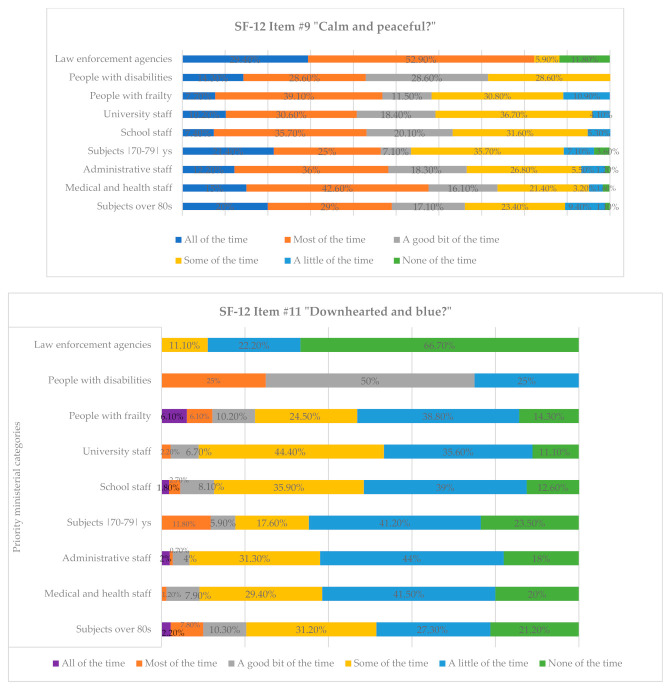
Percentage of response frequencies relative to items #9 and #11 of SF-12 divided by vaccine priority ministerial categories.

**Figure 3 vaccines-09-00612-f003:**
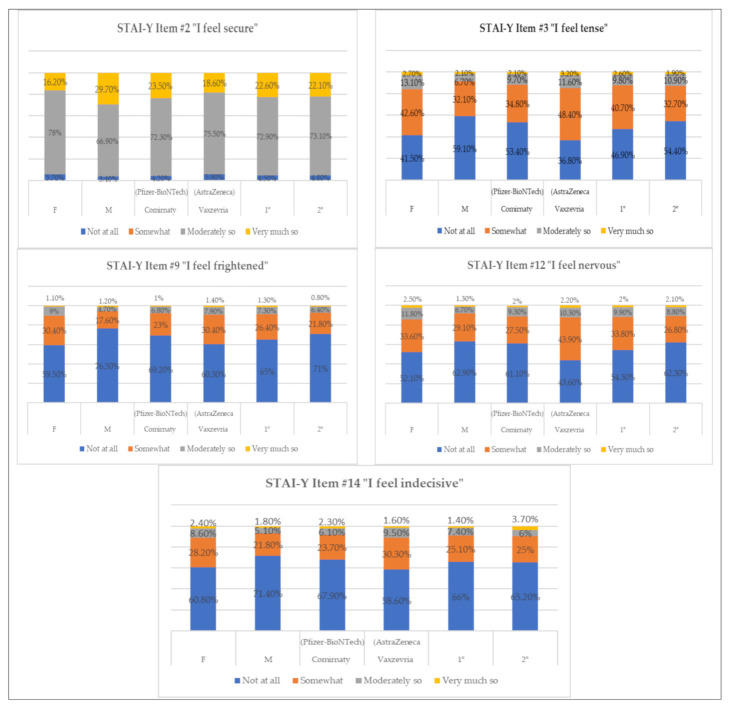
Percentage of response frequencies relative to items of STAI-Y divided by variable.

**Figure 4 vaccines-09-00612-f004:**
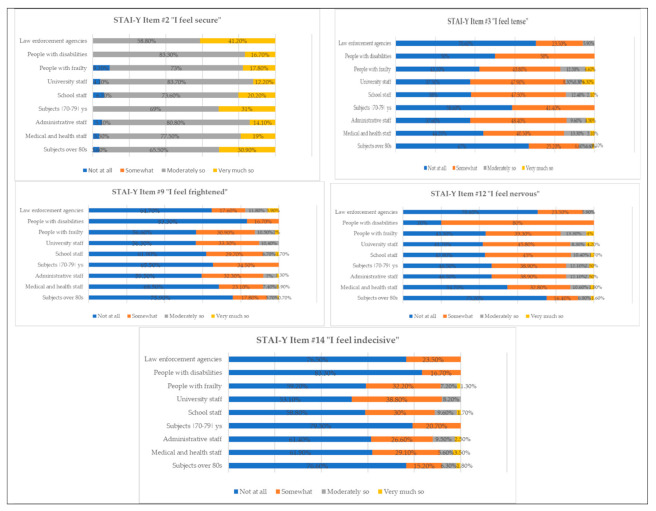
Percentage of response frequencies relative to items of STAI-Y divided by vaccine priority ministerial categories.

**Table 1 vaccines-09-00612-t001:** Frequencies in percentage (%), mean age (M) and standard deviation (SD) of the characteristics of the sample of participants.

	%	M	SD
**F**	54.2	55.88	71.14
**M**	45.8	56.89	27.67
**1 dose**	65.8	54.21	26.26
**2 dose**	34.2	60.16	86.97
**Cominarty (Pfizer-BioNTech)**	75.7	60.51	62.36
**Vaxzevria (Astrazeneca)**	24.3	42.48	11.91
**Medical and health personnel**	28.1	38.39	27.19
**Administrative staff**	10.6	46.22	11.32
**School staff**	15.6	43.41	11.7
**University staff**	3.1	29.96	3.8
**People with frailty**	10.3	47.45	14.03
**People with disabilities**	0.4	52.43	7.8
**Law enforcement agencies**	1.1	46.82	10.24
**People with advanced age**	28.8	87.73	88.32
**Subjects aged between 70–79 years**	2	74.93	4.61

**Table 2 vaccines-09-00612-t002:** Percentage of response frequencies related to item #9 of the SF-12 divided by variable.

	Sex		Type of Vaccine		Vaccine Dose Comirnaty (Pfizer-BioNTech)	
Calm and Peaceful?	F	M	Comirnaty (Pfizer-BioNTech)	Vaxzevria (AstraZeneca)	1st	2nd
**All of the time**	10.9%	18.6%	15.9%	9.5%	12.3%	18.3%
**Most of the time**	31.6%	41.1%	36.3%	34.4%	36%	35.5%
**A good bit of the time**	17.1%	15.9%	15.7%	19.3%	17.7%	14.5%
**Some of the time**	30.9%	19.3%	24.2%	30.4%	26.9%	23.5%
**A little of the time**	8.6%	3.7%	6.4%	6.1%	6.1%	6.9%
**None of the time**	0.8%	1.4%	1.4%	0.3%	1%	1.3%

**Table 3 vaccines-09-00612-t003:** Percentage of response frequencies related to item #9 of SF-12 divided by vaccine priority ministerial categories.

	Priority Ministerial Categories								
Calm and Peaceful?	Subjects with Advanced Age (over 80s)	Medical and Health Staff	Administrative Staff	Subjects Aged |70–79| Years	School Staff	University Staff	People with Frailty	People with Disabilities	Law Enforcement Agencies
**All of the time**	20%	15%	12.2%	21.4%	7.4%	10.2%	7.7%	14.3%	29.4%
**Most of the time**	29%	42.6%	36%	25%	35.7%	30.6%	39.1%	28.6%	52.9%
**A good bit of the time**	17.1%	16.1%	18.3%	7.1%	20.1%	18.4%	11.5%	28.6%	0%
**Some of the time**	23.4%	21.4%	26.8%	35.7%	31.6%	36.7%	30.8%	28.6%	5.9%
**A little of the time**	9.4%	3.2%	5.5%	7.1%	5.3%	4.1%	10.9%	0%	0%
**None of the time**	1.1%	1.6%	1.2%	3.6%	0%	0%	0%	0%	11.8%

**Table 4 vaccines-09-00612-t004:** Percentage of response frequencies related to item #11 of the SF-12 divided by variable.

	Sex		Type of Vaccine		Vaccine Dose Comirnaty (Pfizer-BioNTech)	
Downhearted and Blue?	F	M	Comirnaty (Pfizer-BioNTech)	Vaxzevria (AstraZeneca)	1	2
**All of the time**	1.9%	1.1%	1.3%	2%	1.7%	1.1%
**Most of the time**	5.7%	1.6%	4.4%	2.2%	3.5%	4.7%
**A good bit of the time**	10.6%	5.9%	8.9%	7%	7.8%	9.7%
**Some of the time**	36.9%	25.1%	30.3%	34.2%	32.8%	27.6%
**A little of the time**	32.1%	41.9%	34.6%	41.2%	37.3%	35.1%
**None of the time**	12.9%	24.4%	20.6%	13.4%	16.8%	21.8%

**Table 5 vaccines-09-00612-t005:** Percentage of response frequencies related to item #11 of the SF-12 divided by priority ministerial category.

	Priority Ministerial Categories								
Downhearted and Blue?	Subjects with Advanced Age (over 80s)	Medical and Health Staff	Administrative Staff	Subjects Aged |70–79| Years	School Staff	University Staff	People with Frailty	People with Disabilities	Law Enforcement Agencies
**All of the time**	2.2%	0%	2%	0%	1.8%	0%	6.1%	0%	0%
**Most of the time**	7.8%	1.2%	0.7%	11.8%	2.7%	2.2%	6.1%	25%	0%
**A good bit of the time**	10.3%	7.9%	4%	5.9%	8.1%	6.7%	10.2%	50%	0%
**Some of the time**	31.2%	29.4%	31.3%	17.6%	35.9%	44.4%	24.5%	0%	11.1%
**A little of the time**	27.3%	41.5%	44%	41.2%	39%	35.6%	38.8%	25%	22.2%
**None of the time**	21.2%	20%	18%	23.5%	12.6%	11.1%	14.3%	0%	66.7%

**Table 6 vaccines-09-00612-t006:** Percentage of response frequencies related to the clinically significant items of STAI-Y divided by variable.

	Sex		Type of Vaccine		Vaccine Dose	
Item #2 I Feel Secure	F	M	Comirnaty (Pfizer-BioNTech)	Vaxzevria (AstraZeneca)	1	2
**Not at all**	5.7%	3.4%	4.2%	5.9%	4.5%	4.8%
**Somewhat**	0%	0%	0%	0%	0%	0%
**Moderately so**	78%	66.9%	72.3%	75.5%	72.9%	73.1%
**Very much so**	16.2%	29.7%	23.5%	18.6%	22.6%	22.1%
**Item #3 I feel tense**	**F**	**M**	**Comirnaty** **(Pfizer-BioNTech)**	**Vaxzevria** **(AstraZeneca)**	**1**	**2**
**Not at all**	41.5%	59.1%	53.4%	36.8%	46.9%	54.4%
**Somewhat**	42.6%	32.1%	34.8%	48.4%	40.7%	32.7%
**Moderately so**	13.1%	6.7%	9.7%	11.6%	9.8%	10.9%
**Very much so**	2.7%	2.1%	2.1%	3.2%	2.6%	1.9%
**Item #9 I feel frightened**	**F**	**M**	**Comirnaty** **(Pfizer-BioNTech)**	**Vaxzevria** **(AstraZeneca)**	**1**	**2**
**Not at all**	59.5%	76.5%	69.2%	60.3%	65%	71%
**Somewhat**	30.4%	17.6%	23%	30.4%	26.4%	21.8%
**Moderately so**	9%	4.7%	6.8%	7.9%	7.3%	6.4%
**Very much so**	1.1%	1.2%	1%	1.4%	1.3%	0.8%
**Item #12 I feel nervous**	**F**	**M**	**Comirnaty** **(Pfizer-BioNTech)**	**Vaxzevria** **(AstraZeneca)**	**1**	**2**
**Not at all**	52.1%	62.9%	61.1%	43.6%	54.3%	62.3%
**Somewhat**	33.6%	29.1%	27.5%	43.9%	33.8%	26.8%
**Moderately so**	11.8%	6.7%	9.3%	10.3%	9.9%	8.8%
**Very much so**	2.5%	1.3%	2%	2.2%	2%	2.1%
**Item #14 I feel indecisive**	**F**	**M**	**Comirnaty** **(Pfizer-BioNTech)**	**Vaxzevria** **(AstraZeneca)**	**1**	**2**
**Not at all**	60.8%	71.4%	67.9%	58.6%	66%	65.2%
**Somewhat**	28.2%	21.8%	23.7%	30.3%	25.1%	25%
**Moderately so**	8.6%	5.1%	6.1%	9.5%	7.4%	6%
**Very much so**	2.4%	1.8%	2.3%	1.6%	1.4%	3.7%

**Table 7 vaccines-09-00612-t007:** Percentage of response frequencies related to the clinically significant items of STAI-Y divided by vaccine priority ministerial categories.

Item #2 I Feel Secure									
	Subjects with Advanced Age (over 80s)	Medical and Health Staff	Administrative Staff	Subjects Aged |70–79| Years	School Staff	University Staff	People with Frailty	People with Disabilities	Law Enforcement Agencies
**Not at all**	3.6%	3.5%	5.1%	0%	6.2%	4.1%	9.2%	0%	0%
**Somewhat**	0%	0%	0%	0%	0%	0%	0%	0%	0%
**Moderately so**	65.5%	77.5%	80.8%	69%	73.6%	83.7%	73%	83.3%	58.8%
**Very much so**	30.9%	19%	14.1%	31%	20.2%	12.2%	17.8%	16.7%	41.2%
**Item #3 I Feel Tense**									
	**Subjects with Advanced Age** **(over 80s)**	**Medical and Health Staff**	**Administrative Staff**	**Subjects Aged** **|70–79| Years**	**School Staff**	**University Staff**	**People with Frailty**	**People with Disabilities**	**Law Enforcement Agencies**
**Not at all**	67%	44.2%	37.6%	58.6%	38%	37.5%	42.1%	50%	70.6%
**Somewhat**	25.2%	40.5%	48.4%	41.4%	47.5%	47.9%	40.8%	50%	23.5%
**Moderately so**	6.6%	13.3%	9.6%	0%	12.4%	8.3%	12.5%	0%	5.9%
**Very much so**	1.1%	2.1%	4.5%	0%	2.1%	6.3%	4.6%	0%	0%
**Item #9 I Feel Frightened**									
	**Subjects with Advanced Age** **(over 80s)**	**Medical and Health Staff**	**Administrative Staff**	**Subjects Aged** **|70–79| Years**	**School Staff**	**University Staff**	**People with Frailty**	**People with Disabilities**	**Law Enforcement Agencies**
**Not at all**	75.9%	68.5%	59.5%	65.5%	61.9%	56.3%	56.6%	83.3%	64.7%
**Somewhat**	17.8%	23.1%	32.3%	34.5%	29.7%	33.3%	30.9%	16.7%	17.6%
**Moderately so**	5.7%	7.4%	7%	0%	6.7%	10.4%	10.5%	0%	11.8%
**Very much so**	0.7%	0.9%	1.3%	0%	1.7%	0%	2%	0%	5.9%
**Item #12 I Feel Nervous**									
	**Subjects with Advanced Age** **(over 80s)**	**Medical and Health Staff**	**Administrative Staff**	**Subjects Aged** **|70–79| Years**	**School Staff**	**University Staff**	**People with Frailty**	**People with Disabilities**	**Law Enforcement Agencies**
**Not at all**	75.2%	54.7%	46.5%	46.5%	42.9%	41.7%	43.3%	20%	70.6%
**Somewhat**	16.4%	32.8%	38.9%	38.9%	45%	45.8%	39.3%	80%	23.5%
**Moderately so**	6.8%	10.6%	12.1%	12.1%	10.4%	8.3%	13.3%	0%	5.9%
**Very much so**	1.6%	1.8%	2.5%	2.5%	1.7%	4.2%	4%	0%	0%
**Item #14 I Feel Indecisive**									
	**Subjects with Advanced Age** **(over 80s)**	**Medical and Health Staff**	**Administrative Staff**	**Subjects Aged** **|70–79| Years**	**School Staff**	**University Staff**	**People with Frailty**	**People with Disabilities**	**Law Enforcement Agencies**
**Not at all**	76.6%	61.9%	61.4%	79.3%	58.8%	53.1%	59.2%	83.3%	76.5%
**Somewhat**	15.2%	29.1%	26.6%	20.7%	30%	38.8%	32.2%	16.7%	23.5%
**Moderately so**	6.3%	5.6%	9.5%	0%	9.6%	8.2%	7.2%	0%	0%
**Very much so**	1.8%	3.5%	2.5%	0%	1.7%	0%	1.3%	0%	0%

**Table 8 vaccines-09-00612-t008:** Mean (M) and standard deviation (SD) of the scores obtained on the PCS and MCS scales of the SF-12 relating to the gender and vaccine category.

	PCS		MCS	
	M	SD	M	SD
F	12.69	1.92	17.40	3.01
M	12.58	2.03	17.23	3.44
Medical and health personnel	12.72	1.88	17.35	3.09
Administrative staff	13.32	1.65	18.34	2.53
School staff	13.16	1.52	18.10	2.34
University staff	13.02	1.46	17.86	2.6
People with frailty	11.61	2.26	14.74	3.72
People with disabilities	10.86	2.47	14.14	4.91
Law enforcement agencies	12.12	1.93	16.0	3.20
People with advanced age	12.48	1.96	17.60	3.12
Subjects aged between 70–79 years	11.23	3.57	15.29	5.86

## Data Availability

Written informed consent has been obtained from the patient(s) to publish this paper.

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
