# Peer review of "Perception of Health, Mistrust, Anxiety, and Indecision in a Group of Italians Vaccinated against COVID-19"

_vaccines, 2021, doi:10.3390/vaccines9060612_

Round 1

Reviewer 1 Report

The analyisis is interesting but purely descriptive. In order to better interpret the results, it would be important to provide more context. In particular as regards the incidence of the epidemic in the area and by age group/professional categories, and as regards general perception on the safety of vaccines. By the latter I refer to wether in national/local media there were reports of problems with the vaccines that might have an influence on the perception of individuals. Given that you do not have time variation, at least the age and "category" dimensions can be illustrated.

Author Response

Dear Reviewer 1,

Thank you for agreeing to review our article and thank you for your suggestions.

point 1 (in yellow)

We have entered data on the context and on the incidence of the epidemic situation in the area and by age groups / professional categories, and on the general perception on the safety of vaccines.

Point 2 (in yellow)

We have inserted any notes on the variation to the variables.

Sincerely,

The authors

Reviewer 2 Report

This is an excellent study and well-performed by the authors. The topic of study is of high importance to know for perception of Health, Mistrust, Anxiety, and Indecision in a Group of Italians Vaccinated against COVID-19. As a reader it was enjoyable to read the article and as a reviewer I could not suggest any modifications in the content.

Author Response

Dear Reviewer 2,

 Thank you for agreeing to review our article and thank you for your positive review.

Sincerely,

 The authors

Reviewer 3 Report

The article could be interesting but some revisions are needed. 

1. Data about the first objective would be better appreciate if they were expressed in histogram and even data about the total sample reported scores of the PCS and MCS could be show. 

2. The second objective of the study (the correlation analysis) in terms of   Statistical Analysis, results description and discussion is not clear. It would be useful to develop that part, to better explain the type of analysis and, if possible, to add tables or figures that help to understand the results and to discuss them more deeply.  

Author Response

Dear Reviewer 3,

Thank you for agreeing to review our article and thank you for your suggestions.

1)  We have expressed in histogram  the data and we reported scores of the PCS and MCS for the object 1 (in red)

  1. We have better explain the type of analysis and we have add tables and graphics for the object 2 (in red)

Sincerely,

The authors

Reviewer 4 Report

Dear authors,
From my point of view this study is relevant and shows interesting news. The study design is adequate. The manuscript is well structured and comprehensive, all relevant information is included. The results are clearly presented. The conclusions are plausible and support the results. I did not notice any errors in the English language. The references seem correct and necessary to me.

Author Response

Dear Reviewer 4,

 Thank you for agreeing to review our article and thank you for your positive review.

Sincerely,

 The authors

Reviewer 5 Report

Journal: Vaccines (ISSN 2076-393X)

Manuscript ID: vaccines-1236136

Title: Perception of Health, Mistrust, Anxiety, and Indecision in a Group of Italians Vaccinated against COVID-19

Comments to the authors:

Summary: The paper evaluates the psychological factors of health perception, mistrust, anxiety, fears, and indecision of Italians vaccinated against COVID-19.

The authors conducted an analysis of the relationships between the psychological factors and other variables: sex, vaccine priority ministerial categories, and the type and dose of vaccine. The findings state that these factors were influenced by some variables, such as the type of vaccine administered, age, and whether respondents fell into a professional role of care and safety of other people. I recommend the publication of this article after consideration of a few minor comments below.

  1. The authors should check for punctuation and spelling errors (line 69, 88, 332).
  2. Would it be possible for the authors to elaborate the rationale and the significance of this study in a bit more detail?
  3. Can the authors justify about the collected data sets given the diversity of the participating subjects in terms of age, sex and categories.
  4. Authors have mentioned that differences emerged for the type and time of vaccine administration. Can they comment if they have included a reason for these differences in the questionnaire.
  5. Can the authors comment on how their findings compare with global surveys with much more larger participants in similar studies.
  6. Can the authors elaborate on the significance of this study with socio economic status of the participants being different in different parts of the country as well as the entire world.

Author Response

Dear Reviewer 5,

Thank you for agreeing to review our article and thank you for your suggestions.

  1. We have checked the punctuation and spelling errors in the total paper, with a MDPI Language editing service.
  2. We have elaborate the rationale and the significance of the study in a bit more detail. (in green)
  3. We have justify about the collected data sets for   the participating subjects in terms of age, sex and categories.(in yellow)
  4. We have specify the reason for the differences emerged for the type and time of vaccine administration. (in yellow)
  5. We have add three global surveys with much more larger participants. (in green)
  6. We have specify the socio economic status question of the participants. (in green)
